# An Optimal Choice of Profile Shift Coefficients for Spur Gears

**Zoltán Tomori**

Institute of Machine Tools and Mechatronics, University of Miskolc, 3515 Miskolc, Hungary; tomori.zoltan@uni-miskolc.hu

**Abstract:** Methods used earlier for choosing profile shift coefficients were based on presumed advantages without physical proof justifying them. In this paper, a new method is proposed which guarantees a positive influence on gear failures and the operational conditions of gear pairs. The author proposes the introduction of a new concept: the cumulative effectiveness of profile shift coefficients.

**Keywords:** profile shift coefficient; gear failures; optimal gear operations; specific sliding; sliding velocity; Almen product; bending stress; cumulative effectiveness of a profile shift coefficient

## 1. Introduction

In the case of the use of spur gears, the required movement ratio must be carried out so that the structural elements are able to transmit the loads occurring without damage. The legality of the movement—that is, the gear ratio—is defined as the choice of the number of teeth of the gears. The proper load capacity is determined by the module, the center distance, and the face width based on the strength considerations. Recording of the listed data is often influenced by the space available for installation.

In the case of spur gears, the knowledge of the numbers of teeth, the module, and the center distance, as well as the pressure angle, determine the working pressure angle, which generally differs from the pressure angle. It follows from the criterion equation of the backlash-free connection that the sum of the profile shift coefficients must be provided to achieve a given working pressure angle. While the profile shift coefficients of the gears are, in principle, infinitely many kinds, in practice they can be chosen at least in several ways. It is only necessary to ensure that the sum of the two profile shift coefficients satisfies the condition of a backlash-free connection.

Although the previous statement is correct, with the exception of some special cases (the very precise ratio of rotations, kinematic drive for accurate angular motion, rotary table for precise positioning of the machine tools, actuator drives, etc.), there is always a need for some backlash.

After determining the sum of the profile shift coefficients, the determination of the profile shift coefficient for the associated gears is one of the "mystical" areas of the gear design. In the sense of the fact that the literature knows quite a number of arbitrary or real reasons for the best possible, but at least very good method for determining the paired $x_1 - x_2$ pair of values. These various gear design systems, after their appearance, are running a very diverse lifestyle, in some way losing their popularity, confirming the principle that every principle of working and loading conditions is equally ideal do not exist, as in other areas of life.

In this paper, the author reviews the general and applied procedures for determining the profile shift coefficients. We show whether there are any constraints in choosing or defining these values.

We also examine what kind of positive effect we have on the operation of the spur gear pair by fulfilling a particular criterion (mode of stress or operating characteristic).

It is considered to be a beneficial effect if we contribute to avoiding a form of damage to the gear by using profile shift coefficients satisfying the criteria, or by improving the

operating conditions such as, e.g., we provide better lubrication conditions, reduce friction losses, and increase efficiency.

After the evaluation, the author proposes a new method for selecting the profile shift coefficients, which will result in an optimal solution for each form of damage or operational characteristic.

The defined profile shift coefficients are considered to be optimal if they provide the best result for a particular criterion. The latter may be the maximum or minimum of a given characteristic.

The author has introduced the concept of cumulative effectiveness of profile shift coefficients for further testing.

The group of criteria functions examined from the point of view of cumulative effectiveness should be compiled from the stress functions critical to the given operating conditions or from the features relevant to the operation.

## 2. The Operating Characteristics of Spur Gears. The Most Commonly Used Methods for Choosing the Profile Shift Coefficients (Equalizations) and Their Critical Analysis

One of the basic problems with the operation of spur gears is that, apart from the pitch point of the gear profile, the tooth profiles in meshing are sliding. The pitch point is the only point on the tooth profiles in meshing where a "clean" (slip free) roll is realized. This operating characteristic, as well as the steep change in load, along the path of contact, together cause the stresses of the gears to be used and their failures.

Knowing the sum of profile shift coefficients, it is customary to use several principles to calculate the profile shift coefficients of the gears in meshing; e.g., based on the equalization of the specific sliding values at the endpoints of the path of contact, or on the same characteristic points, on the basis of equalizing the value of the flash temperatures, etc.

It is important to clarify when selecting the profile shift coefficients whether their value has a lower and upper limit and, if so, which one.

In the case of spur gears, a geometrically correct, non-interfering tooth profile generation is possible between the limit values of tooth profile shift coefficient (e.g., the limits are established by the undercut and the pointing) that causes tooth failures. It is important to note that the minimum value of the contact ratio can reduce the allowable profile shift coefficient range.

The most commonly used methods for selecting profile shift coefficients are the so-called equalizations, which are the equalizations of the values of a tooth characteristic in two characteristic points on the path of contact.

Equalizing the sliding velocities means making their values, NIEMANN's suggestion, equal to each other at the boundaries of the path of contact [1].

Equalizing the specific slidings means making their values, DIKER's suggestion, equal to each other at the boundaries of the path of contact [2,3].

Concerning the mentioned methods, the kinematic conditions of the movement of the teeth in meshing are taken into consideration exclusively regardless of the size and nature of the load. The load does not play any role in the calculations. Based on this, there is no correlation between avoiding damage, good efficiency, and the mentioned gear characteristics.

The problem of ignoring loads is eliminated by balancing the two-factor Almen products [4]. The method is the equalization of the product of the sliding velocity and the resulting Hertzian stress at the endpoints of the path of contact.

Equalizing the Almen product values means making their values, BOTKA's suggestion, equal to each other at the boundaries of the path of contact [5]. In his studies, he showed that if the two-factor Almen products are balanced at the boundaries of the path of contact, then the specific sliding and flash temperature values are also balanced at the same time. This is Botka's three-level equalization principle.

The main problem of the equalization methods is that instead of testing the entire length of the path of contact, only point characteristics are defined [6–9]. This will not ensure that we find the extremum value of the feature we are looking for. Since we do not

know whether the Hertzian stress or the flash temperature at the boundary points of the path of contact has any maximum value at all.

### 3. New Way to Choose Profile Shift Coefficients

#### 3.1. The Stresses of the Teeth

The methods presented above look for the answer to how the sum of the profile shift coefficients $\Sigma x$, which can be calculated by the sum of the number of teeth $\Sigma z$ and the working pressure angle $\alpha_{wt}$, should be divided between the two gears in meshing. This is not a problem for the $u = 1$ gear ratio, while $x_1 = x_2 = 0.5 \times \Sigma x$ because of $z_1 = z_2$. However, at a higher gear ratio, the method of dividing $\Sigma x$ has a greater influence on the properties of the gear in meshing. In order to judge the influence of the division of $\Sigma x$ on gears in meshing, it is expedient to examine the known modes of failure of the gears. It is also worth mentioning the construction methods that can be used to prevent any form of damage. In other words, it is important to know the geometrically achievable load limits for a given gear pair at an early stage of design. If we select from the different types of failures those that can be influenced by determining the size of the tooth, i.e., the design of the tooth, then the causes of these can be classified into the following four groups:

- tooth breakage, caused by the bending stress at the root of the tooth,
- pitting, caused by the compressive stress of tooth surfaces,
- scuffing or scoring caused by the flash temperature on the tooth surfaces,
- wear of the tooth surface, caused by insufficient lubrication.

Depending on the operating conditions (load, circumferential speed, temperature, vibrations, dynamic effects, etc.), the cause of the failure in the case under investigation is the most dangerous for the smooth operation of the gear unit.

#### 3.2. The Limits of the Profile Shift Coefficients

The range of values that can be taken by the profile shift coefficient of each gear is determined by the interferences that occur during the manufacture of the gears [10].

The lower limit of the range of these recordable values is the undercut during gear manufacturing.

It is a known fact that the starting points of the involute tooth profiles of the spur gear are located on the base circle of the tooth, and the points of the involute tooth profile are located on a circle having a larger diameter than the base circle. An involute profile cannot be created on a circle with a smaller diameter than the base circle. In the tool position where the distance between the working headline and the center of the gear blank is less than the product of the radius of the base circle and the cosine of the pressure angle, the cutting-edge points around the tool head work at a base body location where an involute can no longer be made (see Figure 1). In this case, during the rolling out of the tool, the cutting edges in the vicinity of the gripper remove certain parts of the usable profile section made during rolling in. This phenomenon, which is a kind of production interference, is called an undercut.

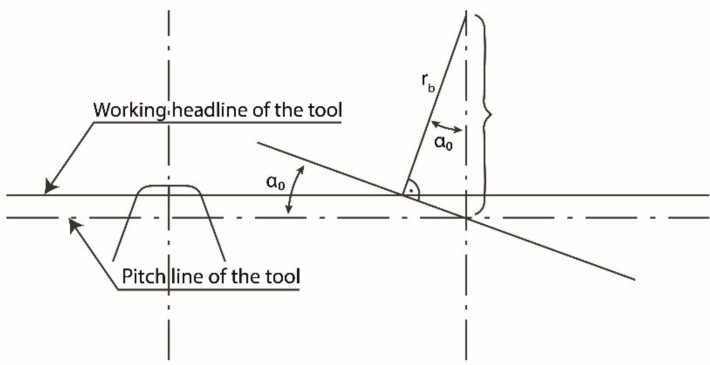

**Figure 1.** Boundary position of the undercut.

The choice of a profile shift coefficients smaller than this value causes an undercut, i.e., the value thus determined is the absolute lower limit of the range providing the correct tooth profile geometry.

The upper limit of the range of values that can be taken is the other production interference that occurs during the production of the gear is called pointing, which means that the tooth thickness at the tip circle or in other words the width of the topland becomes equal to zero.

It is advisable to examine the geometrical conditions in the case of a "pull-out" of the gear cutting tool, i.e., a profile-shift coefficient with a positive value, and whether this "pull-out" has any limit value. By moving the gear cutting tool outwards, i.e., in a direction away from the center of the gear to be toothed, sections of the theoretical evolutionary arc that are further and further away from the base circle are created. In a borderline case, it may be necessary to make the section of the tooth profile where two involute curves of the tooth sides intersecting each other intersect. In this case, during the rolling out of the tool, the cutting edges remove the profile section made at a diameter larger than the circular arc passing through the intersection of the tooth sides. This phenomenon, which is a kind of manufacturing interference, is called pointing. Pointing occurs at the diameter where the topland becomes zero. Thus, the arc section created during rolling in of the cutting tool is cut off when the gear cutting tool is rolled out. Therefore, for diameters larger than this diameter, it is not possible to make an involute for a given gear. In practice, the minimum thickness of the tooth topland must be ensured, because its thinning can cause strength problems.

Choosing a profile shift coefficient higher than this value causes tooth pointing or impermissible thinning of the topland of the gear, i.e., the value thus determined is the absolute upper limit of the range providing the correct tooth profile geometry.

It follows from the above that for each gear to be toothed the range of profile shift coefficients, minimum-maximum value pair, can be determined, where the geometrically correct involute tooth profile can be made. This range is also the usable range of the value of the profile shift coefficient that can be applied for a given gear.

It should also be noted that in the range between the undercut and pointing limits, interference phenomena that interfere with smooth operation may occur, and the value of the contact ratio may not fall below a predetermined minimum value.

*3.3. Selection of Profile Shift Coefficients Based on an Examination Covering the Entire Length of the Path of Contact*

Based on all this, it seems expedient to introduce a new method for the selection of profile shift coefficients, by the application of which the favorable consequences for the teeth stresses can be clearly established [11]. The newly developed method aims to avoid the characteristic damages and to ensure a favorable efficiency by choosing the profile shift coefficients. The examination is extended to the following failures: tooth fracture, pitting, scuffing [12–18], wear [19]. Good efficiency is achieved by minimizing friction loss on the tooth profile and we want to ensure by achieving a favorable lubrication condition [20–22].

The profile shift coefficients chosen in the new method are considered optimal if they provide the best result for a given criterion. The latter can be the maximum or minimum of the given characteristic. The mathematical method of examinations according to different criteria is local extreme value search using the finite increments method in a given interpretation range.

Thus, below the optimal profile shift coefficient, it is defined as $x_1;x_2$ value pair we mean, depending on each test criterion:

- the minimum of the maximums of the characteristic variable varying along the entire length of the path of contact (the bending stress at the root of the tooth, the compressive (Hertzian) stress between the contact tooth surfaces, the magnitude of the resulting flash temperature),

- the maximum of the minimums of the characteristic variable varying along the entire length of the contact line (lubrication film thickness),
- the minimum of the given characteristic, that is, it is sufficient to produce the criterion function in the form $f(x_{1_i})$, then determine the optimal value in the form $f_{opt} = min[f(x_{1_i})]$ (friction loss on the tooth profile, wear).

Using the newly developed method, we divide the range $x_{1min}$; $x_{1max}$ into $n$ equal parts. The interval $\Delta x$ is:

$$\Delta x = \frac{x_{1max} - x_{1min}}{n} \ . \tag{1}$$

The boundaries of the test range can be determined as defined above. The lower limit is indicated by the undercut and the upper limit by the pointing. Choosing the value of $n$ large enough gives a sufficiently dense division.

Within the range boundaries, an intermediate value of $x_1$ is obtained as follows while the value of the step parameter $i$ is:

$$i = 0, 1, \ldots, n \ . \tag{2}$$

Intermediate values of $x_1$:

$$x_{1_i} = x_{1min} + i \times \Delta x \ . \tag{3}$$

For each value of $x_{1_i}$ there is an objective function $f(x_{1_i})$ and along the line of the contact path a critical, e.g., minimum value.

$$f_{min_i} = min[f(x_{1_i})] \ . \tag{4}$$

Examining these values as elements of a vector we can construct a vector containing the minimum values or discrete values of the objective function having $n + 1$ elements. From the elements of the vector, we can select the largest element, which is the maximum of the minima for the given gear in terms of the objective function; that is, the most favorable solution possible.

$$f_{opt} = max(f_{min_i}) \ . \tag{5}$$

The index of the vector element of the objective function with the maximum value gives the value that should be substituted for $i$ to obtain the optimal profile shift coefficient $x_{1opt}$. Knowing $x_{1opt}$, the profile shift coefficient for the other elements of the gear pair can be calculated based on $x_{2opt} = \Sigma x - x_{1opt}$. If the objective function of the investigation is of a nature contrary to that previously described—that is, to find the minimum of the emerging maxima (avoidance of pitting and scuffing), the elements of the vector containing the extreme values produced during the procedure are, respectively, the maximum values belonging to each value of $f(x_{1_i})$, of which the minimum gives the optimal solution and the corresponding $x_{1opt}$ value.

In the third case, it is sufficient to produce the objective function in the form $f(x_{1_i})$, then determine the optimal value in the following form:

$$f_{opt} = min f(x_{1_i}) \ . \tag{6}$$

To demonstrate the use of this method, we examine the distribution of the Hertzian stress along the path of the contact and thus the effect of profile shift coefficients on pitting.

### 3.4. Effect of Profile Shift Coefficients on Pitting

In order to present and evaluate the results of the calculations, it is expedient to record the data of a sample gear pair. By performing the tests according to the different objective functions on this sample gear-pair, the profile shift coefficient value pairs that can be determined on the basis of these principles become comparable.

Data: Number of teeth: $z_1 = 19$, $z_2 = 37$, module: $m = 3$ mm, center distance: $a_w = 86.4$ mm, pressure angle: $\alpha = 20°$, face width: $b = 20$ mm, normal tooth force $F_n = 2500$ N, pinion angular velocity: $\omega_1 = 150$ 1/s, modulus of elasticity of gear materials: $E_1 = E_2 = 206,000$ MPa, Poisson's ratio of gear materials: $\mu_1 = \mu_2 = 0.3$.

The Hertzian stress at each point of the path of the contact, which can cause pitting on the teeth, can be determined by the following line of reasoning.

The Hungarian E. VIDÉKI was the first to propose the Hertzian stress generated during the compression of curved surfaces to calculate the stress on tooth surfaces [23]. The compressive stress due to the load, determined according to Hertz theory, can be determined at any point P of the path of contact as follows:

$$\sigma_H = \frac{2 \times F_n}{\pi \times b \times a_p}, \tag{7}$$

where $a_p$ is the half-width of the contact surface due to compression, $b$ is the tooth width, $F_n$ is the normal tooth force. To perform tests over the entire length of the path of contact, it is advisable to introduce a linear parameter "$l$" as follows.

To keep track of changes along the path of contact, we introduce the linear parameter "$l$", which is interpreted as that its value at the boundary point "$A$" of the path of contact is: $l = 0$, while at the boundary point "$E$" of the path of contact is: $l = \varrho_{1E} - \varrho_{1A} = g_\alpha$. Where $\varrho_{1A}$ and $\varrho_{1E}$ are the instantaneous radii of curvature of the pinion tooth profile at the boundary points, $g_\alpha$ is the contact length. Using the linear parameter "$l$" we can write the values of the radii of curvature at the characteristic points of the path of contact, and we can give the changes of the instantaneous and sum curves along the path of contact.

The radii of curvature at the boundary points of the path of contact are:

$$\varrho_{1A} = a_w \times \sin(\alpha_{wt}) - \rho_{2A}, \tag{8}$$

$$\varrho_{2A} = \sqrt{r_{a2}{}^2 - r_{b2}{}^2}, \tag{9}$$

$$\varrho_{1E} = \sqrt{r_{a1}{}^2 - r_{b1}{}^2}, \tag{10}$$

$$\varrho_{2E} = a_w \times \sin(\alpha_{wt}) - \rho_{1E}, \tag{11}$$

where $a_w$ is the center distance, $\alpha_{wt}$ is the working pressure angle, $r_{a1}$ and $r_{b1}$ are the radii of the addendum, and the base circle of the pinion, $r_{a2}$ and $r_{b2}$ are the radii of the addendum and the base circle of the gear.

The radii of curvature at the boundaries of the individual connection:

$$\varrho_{1B} = \varrho_{1E} - p_b, \tag{12}$$

$$\varrho_{2B} = g_\alpha - \varrho_{1B}, \tag{13}$$

$$\varrho_{1D} = \varrho_{1A} + p_b, \tag{14}$$

$$\varrho_{2D} = g_\alpha - \varrho_{1D}, \tag{15}$$

where $p_b$ is the base pitch.

Values of the instantaneous radii of curvature at any 'P' point of the path of contact:

$$\varrho_1(l) = \varrho_{1A} + l, \tag{16}$$

and

$$\varrho_2(l) = a_w \times \sin(\alpha_{wt}) - \varrho_1(l). \tag{17}$$

While the value of the current sum curve $\Sigma\kappa(l)$ is:

$$\Sigma\kappa(l) = \frac{1}{\varrho_1(l)} + \frac{1}{\varrho_2(l)}. \tag{18}$$

The following load distribution model is used in the tests.
In the connection stages of the two pairs of teeth:

$$Q_{AB}(l) = \frac{F_n}{2} \times \left(1 + \frac{l}{\rho_{1B} - \varrho_{1A}}\right),$$
(19)

and

$$Q_{DE}(l) = \frac{2}{3} \times F_n \left(1 - \frac{\rho_{1A} + l - \rho_{1D}}{2 \times (\rho_{1E} - \rho_{1D})}\right),$$
(20)

while in the connection stage of a pair of teeth:

$$Q_{BD} = F_n .$$
(21)

The load distribution along the path of contact can be seen on Figure 2.

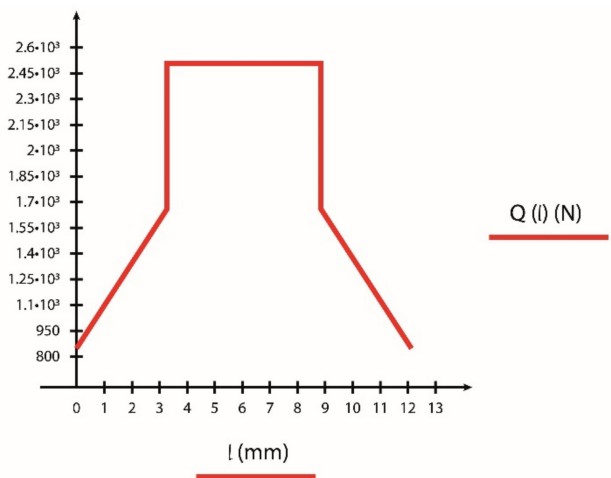

**Figure 2.** Based on the presented load model, the load change along the path of contact.

The half-widths of the contact surfaces can be written that

$$a_p(l) = \sqrt{\frac{4 \times Q(l)}{\pi \times b \times \Sigma\kappa(l) \times E_r}}$$
(22)

where $Q(l)$ is the magnitude of the load along the path of contact, $b$ is the face width, $\Sigma\kappa(l)$ is the current sum curve, $E_r$ is the reduced modulus of elasticity. The value of which can be determined by Equation (23).

$$E_r = \frac{2 \times E_1 \times E_2}{E_1 + E_2}$$
(23)

The change of the instantaneous sum curve $\Sigma\kappa(l)$, along the path of contact, is shown in Figure 3:

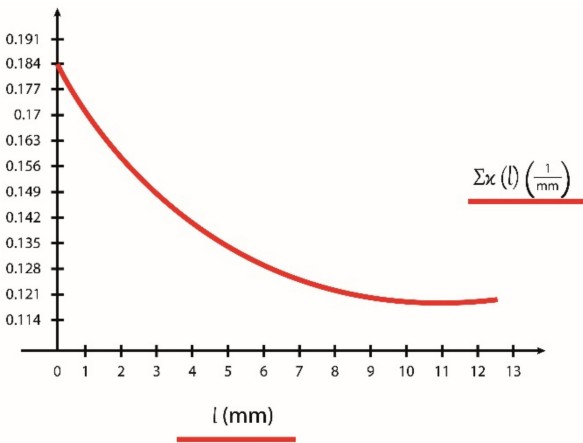

**Figure 3.** The change of the instantaneous sum curve $\Sigma\kappa(l)$, along the path of contact.

Using these, the change in Hertzian stress along the path of contact as a function of the introduced linear parameter $(l)$:

$$\sigma_H(l) = \frac{2 \times Q(l)}{\pi \times b \times a_p(l)} \tag{24}$$

The change in Hertzian stress along the path of contact is shown using the data of the sample example in Figure 4.

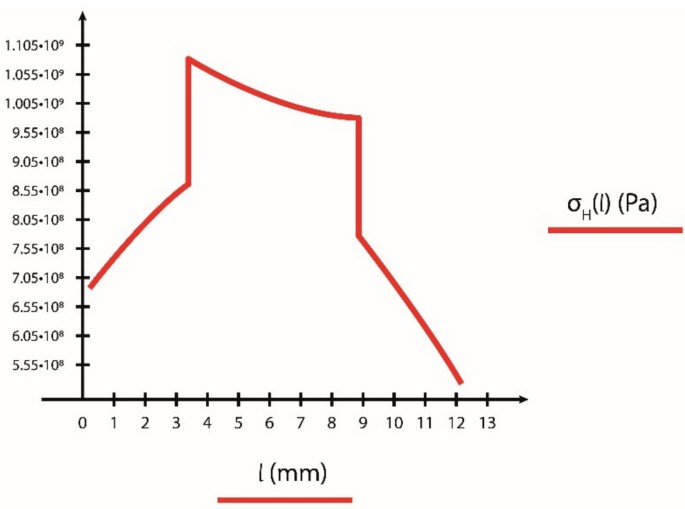

**Figure 4.** The change in Hertzian stress along the path of contact is shown using the data of the sample example.

It can be clearly seen from Figure 4 that the value of the highest Hertzian stress generated is on the inlet side, at the boundary point of the individual connection, for the sample gear. However, its possible location is influenced by the geometrical conditions, so the determination of which point of the connection is dangerous for the Hertzian stress for a given gear can only be decided by a test covering the entire length of the path of contact.

The Hertzian stress in the gear connection is influenced by the profile shift coefficients through the addendum radii of the gear in meshing $r_{a1}$ and $r_{a2}$. Accordingly, their effect is present in the quantities $Q(l)$ and $a_p(l)$ in Equation (24), whereas a change in the profile shift coefficients causes a shift in the AE meshing section along the path of contact, which affects both the load distribution and the size of the Hertzian stress zone.

Based on these, we can examine the change in Hertzian stress along the path of contact at different profile shift coefficient values.

For each value of $x_{1_i}$ performing the test just described, we obtain the objective function $f\left(x_{1_i}\right)$, the set of Hertzian stress functions, from which we can assign to each value of $x_{1_i}$ the extreme value of the corresponding objective function $f\left(x_{1_i}\right)$, in this case the maximum, thus producing a vector containing the maxima of the resulting Hertzian stress functions. The minimum of these vector elements will be the search criteria, in our case to provide the optimal solution for the Hertzian stress $(x_{1opt}, x_{2opt})$. The method should be used similarly for additional stresses (the bending stress at the root of the tooth, the magnitude of the resulting flash temperature, wear) and operating characteristics (friction loss, lubricating film thickness).

The shape of the Hertzian stress along the path of contact shown in Figure 4 clearly demonstrates the expected place of the development of the maximum value of the Hertzian stress which is in the connection stage of a pair of teeth. The place of developing pitting coincides with the place of the maximum Hertzian stress. This result shows satisfactory correlation with referenced literature, see Figures 5 and 6 [24,25].

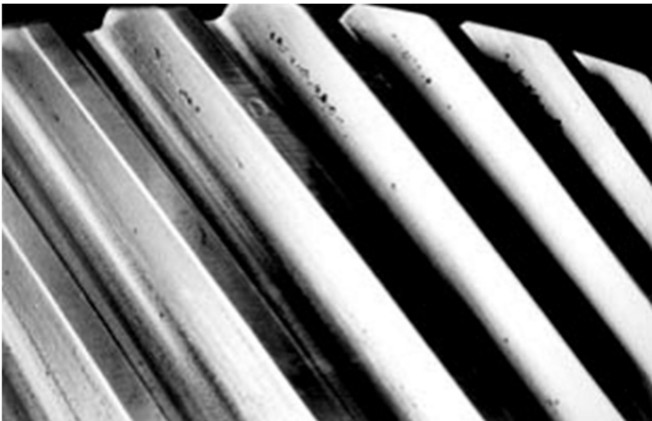

**Figure 5.** The initial pitting in the connection stage of a pair of teeth—1. Reprinted with permission from ref. [24]. Copyright 2003 Springer

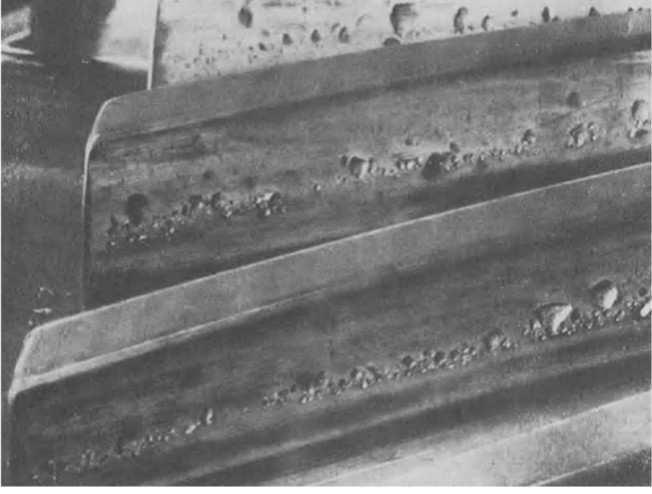

**Figure 6.** The initial pitting in the connection stage of a pair of teeth—2. Reprinted with permission from ref. [25]. Copyright 2002 AGMA

## 4. The Cumulative Effectiveness of a Profile Shift Coefficient

In the sources referenced earlier, e.g., Ref. [2] Diker J.I.; Evolventnije zaceplenyije sz uprjamimi zubcami, Organmetal, 1935. (in Russian), we looked for and named the values of the profile shift coefficients as optimal values, which ensured the avoidance of each form of damage and the achievement of the possible extreme value of various operational

characteristics, separately. That is, the studies so far have only allowed the search for an optimum in some aspect at a time. However, the obvious question is whether there is a value of the profile shift factor $x_1$ that minimizes the risk of multiple damages occurring at the same time, and how the profile shift coefficient, already called optimal in some respects (e.g., $x_1$, which ensures the optimum value of the Hertzian stress value) influences the development of other forms of stress, as well as operational characteristics?

Continuing with the new test method described in detail in point 3, we can arrive at the development of a multi-aspect calculation method that simultaneously examines changes in different forms of stress and operating characteristics along the path of contact [26].

Different stresses and functional characteristics cannot be directly compared due to different physical contents, i.e., units of measure, and different orders of magnitude of formal value. However, it is possible to solve the problem by taking into account the normalized shapes of the different stress and operational characteristic functions. Then it is expedient to generate the normalized function with the maximum value of the set of values belonging to the examined interpretation range because in this way the largest occurring function value will be exactly 1 and all other function values will certainly be even less.

By performing the described normalization on the features to be examined, we can create the sum of the normalized functions of the examined features. These different sum functions can be generated according to which form of damage is considered critical under the given operating conditions, and which of the various examined characteristics of the operation influence the operation of the gear to a significant extent.

It is advisable to introduce the concept of the cumulative effectiveness of profile shift coefficients for further studies. By the cumulative effectiveness of a profile shift coefficient, we mean the position of the sum function of the normalized functions of the various stresses and operating characteristics, how close it is to the best result, i.e., the maximum of the summing. This position of the function can be characterized by defining the area under the function. Thus, the function with the largest area under the function will be the optimal choice for the cumulative effectiveness, the best choice compared to the sum functions formed at the other profile shift coefficient values examined.

Representing the function of the cumulative effectiveness as a bivariate function where the independent variables $(x_1, l)$, a surface is obtained of which plane sections are univariate functions of the cumulative effectiveness. The location of the planar sections is determined by the selected $x_1$ values.

Figure 7 shows the case where the cumulative effectiveness function is formed by the Almen product, the Hertzian stress, and the lubricating film thickness as target functions. The value of $x_1$ takes on the values of the previously defined usable range so that the range between the defined minimum and maximum values is divided into 100 parts, while the path of contact is divided into 30 parts of the range of the linear parameter $l$.

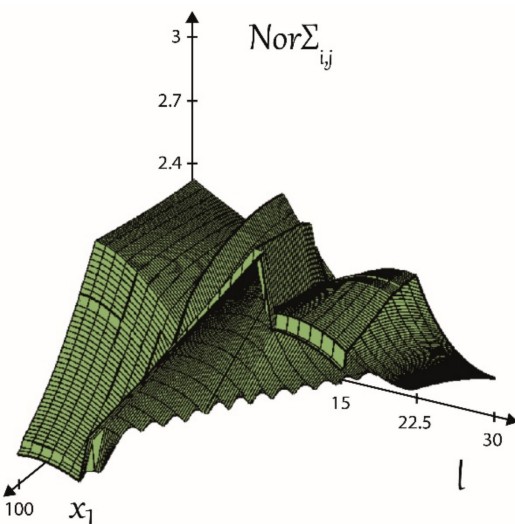

**Figure 7.** The surface of the function of the cumulative effectiveness as a bivariate function.

The function is formed by the Almen product, the Hertzian stress, and the lubricating film thickness as target functions.

## 5. Conclusions

The optimal profile shift coefficient values obtained with the help of the traditional equations (equalizations) and the new method, determined on the basis of various objective functions, are summarized in Table 1, where we first summarized the basic data of the sample gear-pair, then in the results we summarized the $x_{1opt}$ values obtained on the basis of each aspect, and we gave the corresponding $x_{2opt}$ values.

**Table 1.** Results.

| The Data of The Sample Gear Pair: | | |
|---|---|---|
| Number of teeth: | $z_1 = 19$ | $z_2 = 37$ |
| Module: | $m = 3$ mm | |
| Center distance: | $a_w = 86.4$ mm | |
| Working pressure angle: | $\alpha_{wt} = 24$ degree | |
| **Results:** | $x_1$ | $x_2$ |
| Undercut limit on the pinion: | −0.111 | 0.99 |
| Equalization of sliding velocities | 0.349 | 0.53 |
| Equalization of Almen products | 0.479 | 0.4 |
| Equalization of specific sliding | 0.479 | 0.4 |
| Tooth pointing limit of the pinion | 0.943 | −0.064 |
| Minimum friction loss on the tooth profile | 0.374 | 0.505 |
| Optimal bending stress at the root of pinion | 0.943 | −0.064 |
| Optimal Hertzian stress | 0.943 | −0.064 |
| Optimal Almen product | 0.479 | 0.4 |
| Optimal magnitude of the resulting flash temperature | 0.416 | 0.463 |
| Optimal lubricating film thickness | 0.943 | −0.064 |
| Optimal linear wear | 0.943 | −0.064 |

Using the tests performed using the new method, based on the results listed in Table 1, we can reach the following conclusions:

1. The so-called equalization methods (sliding velocity, specific sliding, Almen product, etc.), which determined the profile shift coefficients by equalizing the examined tooth characteristic at the two endpoints of the connection, do not provide an optimal solution according to any of the examined criteria. Neither the avoidance of

the damages examined nor the operating conditions considered provide the most favorable solution.

2.  There is no general solution that is favorable in all respects for the choice of profile shift coefficients. The selection of test criteria, objective functions should always be based on careful consideration of operating conditions and expected damages. However, a choice that satisfies several criteria at the same time is possible, as in our case the largest possible value of $x_1$ is the most favorable solution in terms of bending stress of the at the root of pinion tooth, Hertzian stress, linear wear, and lubricating film thickness.

3.  The effect of profile shift coefficients on tooth damages and operating conditions is through changes in the addendum radii of gears are in meshing. As a result of the change in the radii of the addendum circle of the gears are in meshing, the operating "AE" connecting stage moves along the path of contact, the course of operating characteristics (load, speed, temperature) and the location and magnitude of critical values change.

4.  Changes in the profile shift coefficient of the pinion affect the length of the "AE" connecting stage of the path of contact, as the profile shift coefficient of the pinion increases, the length of the "AE" stage decreases, and at the same time the distance of the boundary point "A" of the connection from the pitch point "C" of the connection is reduced.

5.  The minimum location of the normalized sum functions is at the pitch point of the connection. Moving away from the pitch point, the value of the sum functions increases.

6.  The shape of the normalized sum functions accurately shows the position of the boundaries of the individual connection relative to the pitch point and the boundaries of the connection.

**Funding:** This research received no external funding.

**Institutional Review Board Statement:** Not applicable.

**Informed Consent Statement:** Not applicable.

**Acknowledgments:** Special thanks to András Eleöd, Univesity of Budapest, and József Szente, Univesity of Miskolc, retired university professors for their support providing ideas, encouragement and advise.

**Conflicts of Interest:** The author declares no conflict of interest.

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
