# Peer review of "An Optimal Choice of Profile Shift Coefficients for Spur Gears"

_machines, doi:10.3390/machines9060106_

Round 1

Reviewer 1 Report

The manuscript entitled “An optimal choice of profile shift coefficients for spur gears” is well written, without major typological or grammatical errors. The theme is relevant, the propose of an optimal method to choose the profile shift coefficients for spurs gears is interesting and the application meets the current needs of improvement in this field.

Some comments regarding the paper:

1) The author could give more details on how the optimization shown in equation 4 is done. What method is used for this optimization? What is the stopping criterion?

2) The quality of the figures presented in the paper is not good. The author could improve the quality of the figures.

3) The paper provides analytical analysis only. It should be noted that some kind of experimental validation is necessary for the method, even if this evaluation is made in future works.

Author Response

Dear Reviewer, 

Please see my comments on your points:

1, The detailed description of the stopping criterion and the methodology is desbribed from line 163 to line 170. 

2, Thank you for the comment. The figures will be edited  and improved as requested.

3, Thank you for your note. I shall add a comment on how the theory will be used in future works.

Best regards,
Zoltán Tomori

Reviewer 2 Report

The paper could be interesting, but just before it could be resubmitted again some points are key to make clearer:

The review is complicated by several aspects:

  • State of the art must be improved. There are recent works around the topic that are not mentioned in the state of the art regarding helical or spiral gears. For example, “5-axis double-flank CNC machining of spiral bevel gears via custom-shaped milling tools—Part I: Modeling and simulation” presents an innovation on the way a gear could be manufactured and analyses the difficulties presented in these complex geometries. In the same line, Spiral bevel gears face roughness prediction produced by CNC end milling centers analyses the feasibility for the manufacturing of these components.
  • Additionally, some figures are not in the expected quality for a scientific paper, i.e. Fig. 5
  • Be careful with some of the references: for example, PÁTRIA” irodalmi vállalat és nyomdaipari  this in not common, please translate those titles into English or just update the state of the art. Many of these cases.

Finally, the entire text must be revised, and the performed work must be clearly specified. Here below it is an example of those sentences:

  • “In the previous studies, we looked for and named the values of the profile shift coefficients as optimal values, which ensured the avoidance of each form of damage and the achievement of the possible extreme value of various operational characteristics” Are you referring to this work or to others?

In short, in 2021 spur gears has a lot of tradition and technology is mature. Authors must make extra efforts to present contributions without defects and serious style flaws. Rewrite several parts and submit again.

Author Response

Dear Reviewer, 

Please see my comments below:

State-of-art: Thank you for your comment, however, please note that the paper is not about helic and spiral gear and I am confused by the quoted text which is not from the paper. Please may you explain the correlation between your quote and my paper? 

Figures: Figures will be edited and improved as requested.

References: Thank you for your note. I have requested the correct form of references to be used from the editorial team. It will be corrected asap.

Specification: Duly noted. I shall make it more clear that I am referencing the works published on the topic previously providing examples as well. This and other references will be revised.

Best regards, 
Zoltán Tomori
